# First insight into genetic diversity of two sympatric marten species between the Alps and Adriatic islands

Elena Buzan[1,2]*, Luka Duniš[1], Tilen Komel[1], Boštjan Pokorny[2,3], Carlos Rodríguez Fernandes[4,5], Zoran Marčić[6], Magda Sindičić[7]

**1** Faculty of Mathematics, Natural Sciences and Information Technologies, University of Primorska, Koper, Slovenia, **2** Faculty of Environmental Protection, Velenje, Slovenia, **3** Slovenian Forestry Institute, Ljubljana, Slovenia, **4** CE3C – Centre for Ecology, Evolution and Environmental Changes & CHANGE – Global Change and Sustainability Institute, Departamento de Biologia Animal, Faculdade de Ciências, Universidade de Lisboa, Lisboa, Portugal, **5** Faculdade de Psicologia, Universidade de Lisboa, Lisboa, Portugal, **6** Faculty of Science, University of Zagreb, Zagreb, Croatia, **7** Faculty of Veterinary Medicine, University of Zagreb, Zagreb, Croatia

* elena.buzan@upr.si

## Abstract

Closely related species occupying the same geographical area may exhibit markedly different genetic patterns due to differences in evolutionary history, ecology and behaviour. In this study, a population genetics approach is applied to investigate the genetic structure, diversity, and connectivity of two sympatric carnivore species, i.e., the European pine marten (*Martes martes*) and the stone marten (*Martes foina*) in Croatia and Slovenia. We analysed mitochondrial DNA sequences for both species (28 pine marten and 104 stone marten samples, respectively) and additionally investigated nuclear microsatellite markers for 182 stone martens. For stone marten, we found a significant genetic structuring, with pronounced differentiation between island and mainland populations, and a further substructure within the mainland. But no significant isolation by distance was detected (Mantel test, $p = 0.15$), indicating that differentiation is primarily shaped by island–mainland separation and other geographical discontinuities rather than by distance alone. In contrast, pine marten exhibited moderate haplotype diversity and limited spatial resolution due to the smaller sample size. These contrasting patterns underscore species-specific responses to natural geographical barriers and highlight the need to tailor management strategies accordingly.

## Introduction

Closely related species with different ecological flexibility provide valuable models for understanding how behaviour and habitat use shape their genetic diversity and connectivity among populations. European pine marten (*Martes martes*) and stone

**Data availability statement:** All sequence data submitted to NCBI's GenBank database is also automatically mirrored in the European Nucleotide Archive (ENA) and the DNA Data Bank of Japan (DDBJ). Due to the sensitive nature of the sampling locations collected by citizen scientists (hunters and volunteers), microsatellite genotype data are available from the Laboratory of Molecular Ecology UP FAMNIT on reasonable request at molecular. ecology@upr.si.

**Funding:** This study was funded by: (i) the Slovenian Research and Innovation Agency (programme groups P1–0386 and P4–0107), (ii) the STEPCHANGE European Union's Horizon 2020 Research and Innovation Program under grant agreement No. 101006386, and (iii) The PRO COAST European Union's Horizon Europe Research and Innovation Program under grant agreement No. 101082327. The publication fee was covered by support for researchers provided by the University of Zagreb (M. Sindičić) and the Faculty of Environmental Protection (V4-1285). The funders had no role in study design, data collection and analysis, decision to publish, or preparation of the manuscript.

**Competing interests:** The authors have declared that no competing interests exist.

marten (*Martes foina*) are examples of such species — i.e., coexisting across central and south Europe [1] but differing in adaptability to human-modified landscapes. Understanding their diverse ecological responses can reveal how landscape and habitat fragmentation affect population structure and gene flow; however, only few studies have analysed their genetics in areas of sympatry [2].

Pine marten and stone marten are morphologically similar medium-sized mustelids. Across Europe, stone martens occupy a wide range of habitats—from forests and rural landscapes to urban areas—whereas pine martens remain mainly linked to deciduous, mixed, and coniferous forests [2–7].

Stone marten is generally more adaptable and tolerant to human-modified environments, while pine marten is more strongly associated with forest habitats and therefore considered more sensitive to disturbance [5,8–10]. However, recent observations indicate that also pine martens are increasingly found in rural areas, sometimes excluding stone martens from remaining forest fragments [10–13]. So far, relatively few studies have addressed the genetics of pine and stone martens, mainly focusing on genetic identification and species differentiation in the field [2,14–19] and investigating genetic diversity, population structure, and phylogeographic patterns using nuclear or mitochondrial markers [2,6,20–28].

Both species exhibit complex phylogeographic structures reflecting postglacial recolonization from southern refugia [24,29]. Indeed, previous studies identified several glacial refugia in the Balkans and Anatolia, showing that European populations expanded from these regions after the Last Glacial Maximum [22–27]. Stone marten lineages from eastern and western Anatolia, as well as the Balkans, reveal historical gene flow across Europe and limited regional differentiation, while high mitochondrial diversity in Greece suggests an additional refugial area in southeastern Europe [22–24,27].

Microsatellite studies of the two species have provided important insights into their genetic diversity, structure, and dispersal at regional scales. Wereszczuk et al. [21] demonstrated that pine marten populations in Poland expanded north-eastward following environmental changes and climate warming. Other microsatellite-based studies across Europe revealed moderate population structuring, often shaped by rivers, mountains, and anthropogenic barriers such as roads and urban areas [20,30,31]. These findings underline how both natural and human-induced landscape features influence marten connectivity and local differentiation, providing an important context for evaluating populations in the Dinaric and Adriatic regions.

The Dinaric Mountains, which stretch along the Adriatic Sea from Slovenia to Albania, represent one of the most extensive and contiguous forest landscapes in Europe, providing crucial habitat for both pine and stone marten. In Slovenia, the Dinaric range merges with the Prealpine and Alpine foothills, forming a continuous ecological corridor that supports connectivity among populations of the species. Pine marten is widespread in these interconnected regions, including the Dinaric, Alpine, and Prealpine zones, consistent with its occurrence in the neighbouring Friuli-Venezia Giulia region of Italy. In contrast, stone martens are more common in sub-Mediterranean areas of Slovenia—where pine martens are largely absent—and

typically inhabit mixed forests [32] as well as rural and (sub)urban areas. Although ecological differences exist between the two species, they may occur sympatrically in some transitional zones [32]. In Croatia both species occur across continental and Mediterranean regions, including many Adriatic islands. Although populations of both species are considered stable and legally managed as game species in both Croatia and Slovenia, reliable data on population size, demographic trends, and fine-scale distribution remain scarce [33].

The primary aim of our study was to assess the genetic diversity of pine and stone martens in Croatia and Slovenia. In particular, we aimed to study the spatial distribution of genetic lineages using mitochondrial DNA control region sequences for both species. Due to the limited number of reliably identified pine marten samples—mainly due to difficulties in morphological identification by non-expert collectors—we focused the analysis of population structure using microsatellite markers on stone marten, for which a sufficient number of samples was available. Nevertheless, pine martens were also included in the study to characterize mitochondrial haplotype diversity and to assess interspecific genetic differentiation.

## Materials and methods

### Samples and DNA extraction

A total of 182 stone marten and 28 pine marten samples from Croatia and Slovenia were included in the analysis (Fig 1 and S1 Table). Muscle samples were collected from roadkill, naturally deceased or hunted animals, by researchers, hunters and volunteers. No animal was shot or otherwise killed for the purposes of this study. The sampling procedures were reviewed by the Committee for Equality, Equal Opportunities and Bioethics, Faculty of Environmental Protection, Velenje, Slovenia, which approved an exemption from an ethical opinion/approval (Ref. CEEOB FVO), noting that the work does not fall under legislation on animals used for scientific purposes because procedures were conducted post mortem and animal-welfare standards were respected. Hunting was carried out in accordance with Council Regulation (EC) No. 1099/2009 and relevant national hunting regulations and ethical standards. Tissue samples (2 × 2 mm) were preserved in 70% ethanol, and before the DNA extraction they were air-dried under sterile conditions to remove the ethanol. The DNA was extracted using the Qiagen QIAamp® DNA kit according to the manufacturer's instructions (Qiagen, Hilden, Germany). The DNA was eluted in the appropriate elution buffer and the sample concentration was measured with Qubit 3.0 using the Qubit dsDNA BR Assay Kit (Thermo Fisher Scientific, Waltham, MA, USA). The purity of isolated DNA was also determined using the Epoch™ spectrophotometer (BioTeck, Winooski, VT, USA) measuring the 260/280 and 260/230 absorbance ratio. The extracted DNA was stored in the freezer at −20°C until further analysis.

### Mitochondrial control region amplification and sequencing

A partial fragment of the mitochondrial control region (CR) was amplified using the versatile primer set Dloop-MelR: 5′-ATGTCCTGTAACCATTGACTG −3′ [35] and LRCB1: 5′- TGGTCTTGTAAACCAAAAATGG −3′ [36]. All polymerase chain reactions (PCR) were performed in a 20 μl reaction mix, using DreamTaq Green PCR Master Mix (Thermo Fisher Scientific) and amplified on a Thermal Cycler 2720 (Applied Biosystems). Amplification was performed under the following conditions: incubation for 3 min at 95°C, 30 cycles with 30 s at 95°C, 45 s at 61°C and 60 s at 72°C, followed by a final extension step of 10 min at 72°C. Afterwards, 3 μL of PCR product was purified using a mix consisting of 0.2 μL of exonuclease (Exo I) (Thermo Fisher Scientific), 1 μL alkaline phosphatase (Thermo Fisher Scientific) and 2.8 μL DreamTaq Green PCR Master Mix (Thermo Fisher Scientific). The purification reaction was performed under the following conditions: 30 min at 37°C followed by 15 min at 80°C. Next, two Sanger sequencing reactions were performed in which 3 μL of purified PCR product was sequenced using a mix of 4.2 μL PCR grade water, 1.5 μL BD 5x Buffer (BD biosciences, Becton Drive Franklin Lakes, NY, USA), 1 μL of BigDye Terminator v3.1 sequencing kit (Applied Biosystems) and 0.3 μL of either Dloop-MelR or LRCB1 primer in the concentration of 10μM. The sequencing reactions were performed under the following

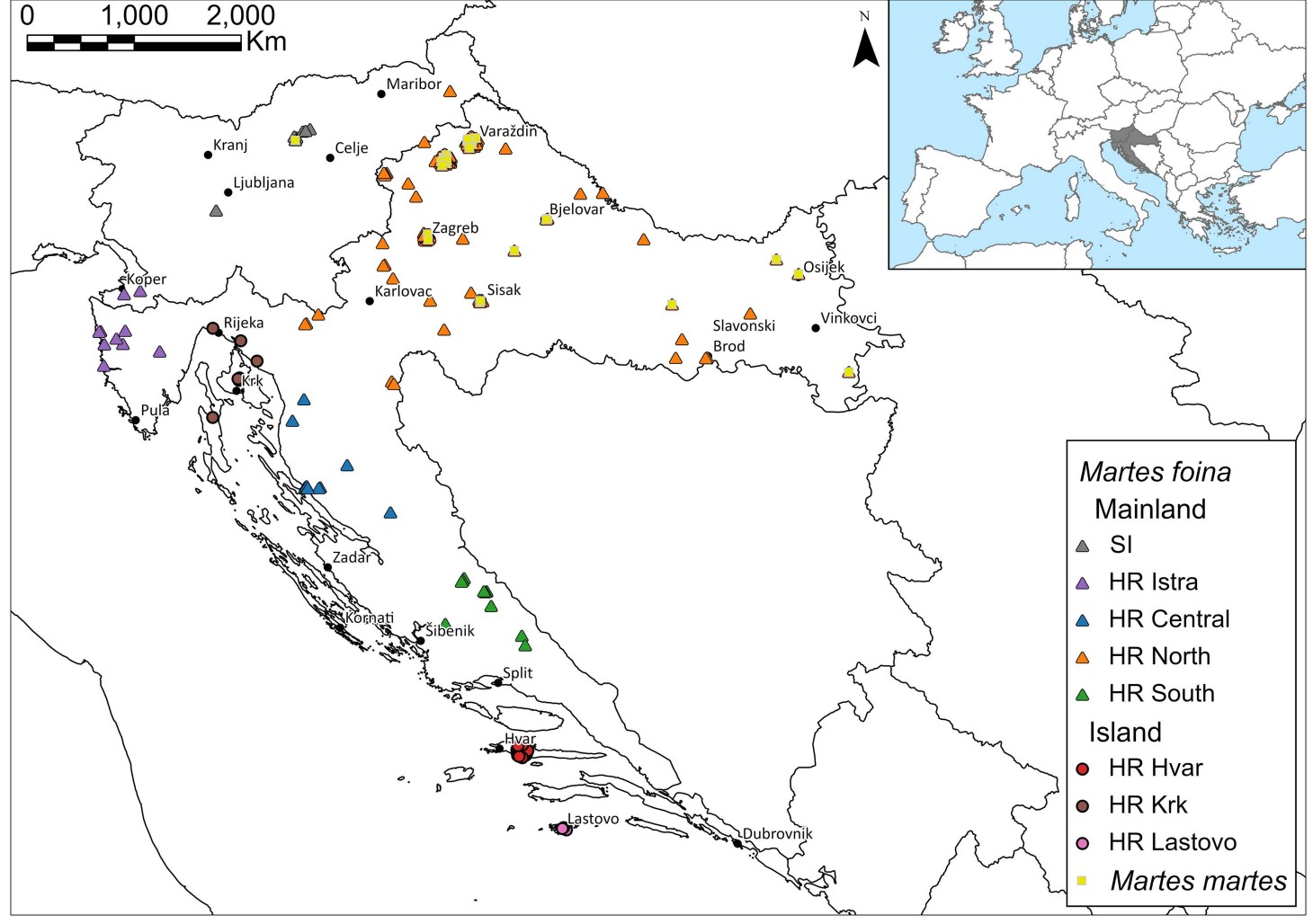

**Fig 1. Sampling localities of stone and pine marten across Slovenia and Croatia.** Each point represents an individual. For *Martes foina*, symbols indicate broad origin (triangles = mainland; circles = islands), and colours indicate sampling location (SI = Slovenia, HR = Croatia): SI (grey), HR Istra (purple), HR North (orange), HR Central (blue), HR South (green), HR Hvar (red), HR Krk (brown), and HR Lastovo (pink). *Martes martes* individuals are shown as squares. The insert in the upper right corner shows the location of the study area within Europe, highlighting the position of Slovenia and Croatia along the north-eastern Adriatic coast. A complete list of sampling sites and sample IDs is provided in Supplementary S1 Table. Country administrative boundaries were downloaded from the geoBoundaries repository [34].

conditions: incubation for 3 min at 96°C, 35 cycles with 10 s at 96°C, 5 s at 50°C and 4 min at 60°C. Electrophoresis of the Sanger sequencing reactions was performed on the Seq Studio Genetic Analyzer (Thermo Fisher Scientific).

### Mitochondrial genetic diversity and population structure analysis

Genetic diversity was assessed with the following parameters: (i) haplotype diversity with standard deviation (Hd ± SD), (ii) nucleotide diversity (π ± SD), (iii) number of haplotypes (h), and (iv) number of polymorphic sites (P). All parameters were estimated with the program DnaSP v.6.12 [37].

The relationship among observed and published haplotypes was evaluated by constructing median-joining haplotype networks (MJN; [38]) using PopART [39]. ArcGIS Pro (ESRI, Redlands, CA, USA) was used for geographical visualisation of data.

Since only a few samples of stone marten from Slovenia were available, the comparison between different geographical areas was conducted at the country level. This was done by calculating the pairwise fixation index (pairwise $F_{ST}$) with 20,000 permutations to measure genetic differentiation [40,41]. Global $F_{ST}$ and pairwise $F_{ST}$ analyses were performed using Arlequin version 3.5.2.2 [42]. To account for different sample sizes, haplotype rarefaction curves were generated in R v4.4.1 [43] using "HaploAccum()" function in the SpideR package [44] with 1,000 permutations. Nonparametric Chao 1 estimator of total haplotype diversity was calculated using the "chaoHaplo()" function. Neutrality tests, including Tajima's *D* and Fu's *Fs*, were calculated in DnaSP v6 to evaluate deviations from neutrality and to infer potential demographic expansions.

## Microsatellite amplification and fragment analysis

The microsatellites were amplified with the ready-to-use KAPA2G Fast Multiplex Mix (Kapa Biosystems, Wilmington, MA, USA), in accordance with the manufacturer's instructions, using 3 μL of template DNA and 0.23 μM of final concentration for each primer used in the set. Amplification was performed under the following conditions: incubation of 3 min at 95°C, 30 cycles with 30 s at 95°C, annealing for 30 s at 55°C, extension for 1 min at 72°C followed by a final extension step of 10 min at 72°C. Fragment analysis was performed on a SeqStudio sequencer (Thermo Fischer scientific) using the GeneScan LIZ500 (−250) standard (Applied Biosystems). The results were validated with GeneMapper v.5.0 software (Applied Biosystems). We amplified 13 microsatellite loci in 3 multiplex sets containing 5, 4 and 4 microsatellites, respectively (S4 Table).

## Analysis of inter-population genetic structure and spatial patterns

The presence of null alleles and heterozygosity deficiency was assessed with FreeNA Version 1.0 [45], which uses the Expectation Maximization (EM) algorithm of Dempster et al. [46] that accounts for deviations from Hardy-Weinberg equilibrium (HWE). Of the 13 microsatellite loci, four were removed from the analysis; Mf 8.8, due to high null allele frequencies; Mf 1.3, Mf 4.10 and Mf 8.10, due to difficulties in PCR conditions standardization and/or inconsistent results on fragment analysis.

Global $F_{ST}$ and pairwise population differentiation ($F_{ST}$) were assessed using Arlequin version 3.5.2.2 [42]. To investigate genetic structuring within the dataset, principal components analysis (PCA) was performed using the ade4 package [47] in R v4.4.1.

For spatial population structure, Geneland v4.9.2 [48] was used within the R environment. This Bayesian method applies MCMC simulations to estimate the number of genetic populations (K) and assigns individuals to clusters based on genetic similarity, incorporating spatial autocorrelation and isolation-by-distance effects.

Finally, isolation by distance (IBD) was tested using Mantel tests, comparing genetic and geographic distances. Euclidean geographic distances were calculated in R using Adegenet [49], and significance was assessed through 999 Monte Carlo permutations, evaluating correlations between Edwards genetic distances and Euclidean distances.

## Results

We amplified 469 bp long mtDNA CR fragments in 104 stone marten samples (97 from Croatia and 7 from Slovenia, which is fewer than the full set used for microsatellite genotyping due to issues with DNA amplification) and 488 bp long mtDNA CR fragments in 28 pine marten samples (27 from Croatia and 1 from Slovenia). The haplotype sequences found in the Croatian samples had been deposited in GenBank already in 2015, but without an accompanying publication and with no metadata on sample locations; this information is provided in S1 Table.

## Mitochondrial DNA genetic diversity of stone marten

A total of 10 haplotypes (labelled as MF1, MF2, MF5, MF6, MF7, MF8, MF9, MF10, MF11, and MF_H25) were identified in the analysed stone marten samples (4 were identified in the Slovenian samples and all 10 in the Croatian samples

(S1 Table)). These were compared with previously published haplotypes from European (Germany, Bulgaria, Greece, Ukraine, Russia, and Turkey) and Asian populations (Turkmenistan, Tajikistan, Russia, Kazakhstan, China, and Pakistan) obtained from GenBank. All haplotypes identified in our samples match previously published ones.

The most frequent haplotype among the analysed samples was MF2 (25.2%), followed by MF7 (24.3%) and MF11 (19.6%) (S1 Table). The haplotypes MF1, MF6, MF8, MF9, MF10, and MF_H25 were detected exclusively in Croatia, whereas MF2, MF5, MF7, and MF11 were found in both countries.

The haplotype accumulation curves by population (S1 Fig) did not reach an asymptote, suggesting that further sampling is likely to increase the haplotype diversity. The value of the haplotype accumulation curve for Croatia at seven individuals is 4.2 haplotypes, while the total number of haplotypes discovered in Slovenia in seven analysed samples is four, indicating that the inclusion of additional Slovenian samples would likely reveal more haplotypes. The estimated Chao1 richness indicated that approximately 14.5 haplotypes may be present in Croatia and 6.0 haplotypes in Slovenia. The lower Chao1 estimate for Slovenia is probably a consequence of the limited number of analysed sequences ($n = 7$).

The total number of polymorphic sites was nine. The overall haplotype diversity was $0.811 \pm 0.015$ and nucleotide diversity was $0.005 \pm 0.001$. The Croatian group showed higher nucleotide ($\pi = 0.006 \pm 0.001$) and haplotype diversity (Hd $= 0.814 \pm 0.015$) compared to the Slovenian group ($\pi = 0.005 \pm 0.002$ and Hd $= 0.810 \pm 0.039$, respectively) (S5 Table) but the possible reason for this result is the limited sample size in Slovenia, which is also reflected in the lower estimate of Chao 1 richness and the lower number of total haplotypes observed. Tajima's D test yielded a positive value for the Croatian group, which may suggest recent population expansion, and a negative value for the Slovenian group, potentially indicating genetic drift. However, as none of the results were statistically significant, definitive conclusions cannot be drawn—particularly for the Slovenian population, where the small sample size limits interpretative power.

## Insight into phylogeographic structure of stone marten from mitochondrial haplotype network

A total of 85 stone marten mtDNA CR sequences representing unique haplotypes were included in the median-joining network (S2 Table). After trimming all sequences to the overlapping 469 bp fragment, this number was reduced to 47 haplotypes. The network suggested two major haplogroups with a clear geographical division (Fig 2). The cluster containing haplotypes from Eastern Turkey also includes four haplotypes from Croatia (MF6, MF8, MF9, and MF10), along with the common haplotypes MF_H1 and MF_H15, which is shared by samples from Eastern Turkey and Slovenia. Within the Major Eurasian group, the majority of Croatian samples belong to the MF-H26 haplotype, while the remaining samples are distributed among MF-1, MF_Hap6, and MF_Hap9. A smaller number of samples belong to haplotype MF-Hap6, which is also shared with samples from Greece and Bulgaria. Samples from Slovenia were assigned to previously published haplotypes MF-1, MF-Hap9, and MF_H26. Interestingly, MF-1 and MF-Hap9 also include samples from Western (i.e., European part of Russia) and Western Ukraine despite these regions being geographically distant from the current study area. The MF_H26 haplotype has a wider distribution, occurring in southern Germany, Bulgaria, Greece, and western Turkey. The star-like topology in the Major Eurasian group is observed around the common haplotype MF_H26, with most haplotypes closely related by a single mutation step. The intermediate haplotype MF-1 also exhibits a slightly weaker star-like structure, with four mutation steps connecting it to the Eastern Turkey cluster and three mutational steps from the central haplotypes in the second cluster MF_H26.

The overall haplotype diversity was high, while nucleotide diversity was relatively low (Table 1). Both the Major Eurasian and Eastern Turkey groups showed slightly lower haplotype diversity but comparable nucleotide diversity. Tajima's D values were not statistically significant in any group, whereas Fu's Fs values were strongly negative and significant, indicating population expansion.

## Spatial genetic structure of stone marten

The principal component analysis based on individual multilocus genotypes revealed clear genetic structuring among stone marten populations from Croatia and Slovenia (Fig 3). The first two principal components explained 49.7% and

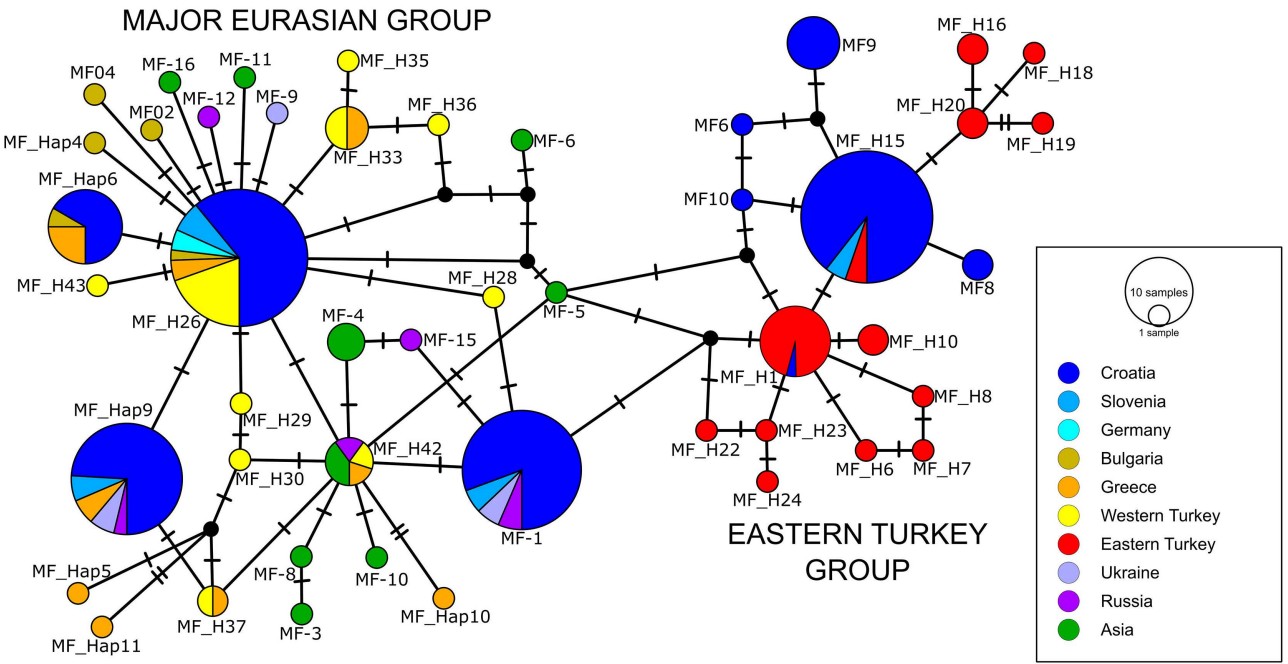

**Fig 2. Median-joining haplotype network for 85 mtDNA control region sequences (469 bp) of stone marten (S1 and S2 Tables).** Each circle indicates a haplotype (the name is given next to the circle), with the size of the circle proportional to the number of individuals bearing that haplotype. Slash marks in lines connecting the circles indicate the number of nucleotide substitutions between adjacent haplotypes. Black dots indicate intermediate haplotypes not detected. The colour of each circle indicates the country or geographical region (in the case of Turkey) where each haplotype was found. We used the same colour for these regions as Ishii et al. [24].

**Table 1. Molecular diversity indices based on mitochondrial control-region sequences of stone martens generated in this study and retrieved from GenBank, trimmed to the overlapping 469 bp fragment.**

|  | P | h | Hd (SD) | π (SD) | Tajima's D | Fu's Fs |
|---|---|---|---|---|---|---|
| Overall (7299) | 34 | 47 | 0.919 (0.008) | 0.007 (0.001) | −1.123 (p>0.10) | −32.055 (p<0.001) |
| Major Eurasian group (n=210) | 26 | 30 | 0.860 (0.015) | 0.004 (0.001) | 1.791 (p>0.05) | −22.643 (p<0.001) |
| Eastern Turkey group (n=89) | 16 | 17 | 0.864 (0.019) | 0.004 (0.001) | −0.980 (p>0.10) | −6.786 (p=0.001) |

P: number of polymorphic sites, h: number of haplotypes, Hd: haplotype diversity, π: nucleotide diversity, SD: standard deviation.

13.1% of the total genetic variation, respectively. Individuals from the island of Hvar (HR Hvar) formed a distinct and well-separated cluster along the first axis, indicating pronounced genetic differentiation from the mainland populations. Samples from islands Krk and Lastovo were partly separated from the mainland cluster but showed some overlap, suggesting moderate levels of gene flow or shared ancestry. In contrast, individuals from Slovenia and the Croatian mainland occupied a central, more diffuse region of the plot, consistent with genetic admixture or continuous population structure.

The spatial genetic structure inferred from the Geneland analysis identified four distinct genetic clusters, each showing strong geographical association (Fig 4). These spatial patterns were broadly consistent with the PCA results, where island populations—particularly those from Hvar—formed clearly separated genetic groups, while mainland populations exhibited more admixture. The overall pattern supports limited gene exchange between island and mainland populations and highlights the strong divergence of the Hvar island population, likely resulting from long-term isolation and genetic drift. Clusters 1 and 2 of the Geneland output largely correspond to individuals from mainland regions, particularly those assigned to admixed or overlapping clusters in the PCA plot. Cluster 3, whose high posterior probabilities are centred on

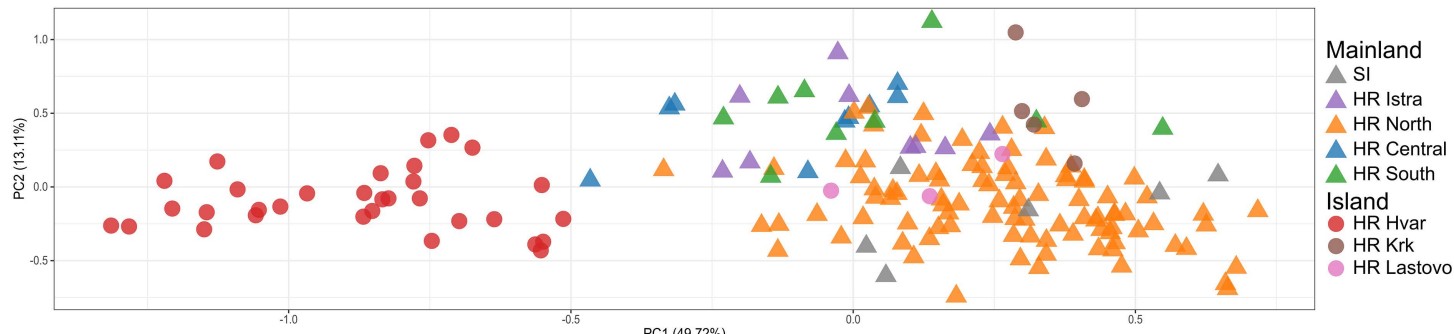

**Fig 3. Analysis of population structure of stone marten in Croatia and Slovenia using PCA.** The main plot shows the first two principal components (explaining 49.7% and 13.1% of the variation, respectively). Each point represents an individual, with symbols indicating broad origin (triangles = mainland, circles = islands) and colours indicating sampling location; colour codes follow the description in Fig 1.

the southern region, likely reflects a geographically isolated population—possibly corresponding to individuals from or near the southern Adriatic islands (e.g., Lastovo island). Cluster 4, which spans northern and central areas, includes individuals that formed intermediate groupings in the PCA space.

The results of the pairwise $F_{ST}$ analysis (Table 2) show a very high genetic differentiation between the samples from the island of Hvar and the rest of the samples, with the highest differentiation with the samples from the island of Lastovo (0.546, p < 0.05). The samples from the island of Lastovo also showed high genetic differentiation with the samples from the mainland.

Microsatellite-based genetic distances (S2 Fig) were not correlated with geographical distances among populations (observation = 0.235, p = 0.133).

## Mitochondrial DNA genetic diversity of pine marten

The alignment of partial mitochondrial control region sequences (488 bp) from 28 pine martens—one from Slovenia and 27 from Croatia—revealed eight haplotypes. Four of these haplotypes (Mm19, Mm5, Mm43, and Mm44) had been previously described by Ruiz-González et al. [29]., while the remaining four (MM2, MM3, MM5, and MM6) were deposited in GenBank in 2015 without an accompanying metadata. Due to the limited sample size—comprising only one individual from Slovenia and 27 from Croatia—we assessed genetic diversity across the combined sample set rather than by country. The analysis revealed a total of nine variable sites, representing 1.8% of the total 488 bp mitochondrial control region sequence. Overall haplotype diversity was 0.828 ± 0.042, and nucleotide diversity was 0.007 ± 0.001, indicating moderate mitochondrial genetic diversity within the sampled individuals.

## Insight into phylogeographic structure of pine marten from mitochondrial haplotype network

A median-joining haplotype network was constructed using a total of 161 haplotypes (Fig 5), including both newly generated sequences and those retrieved from GenBank (S1 and S2 Tables). Haplotypes previously deposited in GenBank were named according to their original designations (S2 Table). The network analysis supports the phylogenetic structure described by Ruiz-González et al., identifying two major haplogroups within *M. martes*: the Fennoscandian–Russian (FNR) group and a broader cluster comprising nearly the entire European range of the species [29]. This latter group is further subdivided into two phylogeographic lineages: the Mediterranean (MED) and the Central–Northern European (CNE) groups. Our samples were assigned to both groups; however, all haplotypes identified in this study and deposited in 2015 in GenBank fall into the CNE lineage.

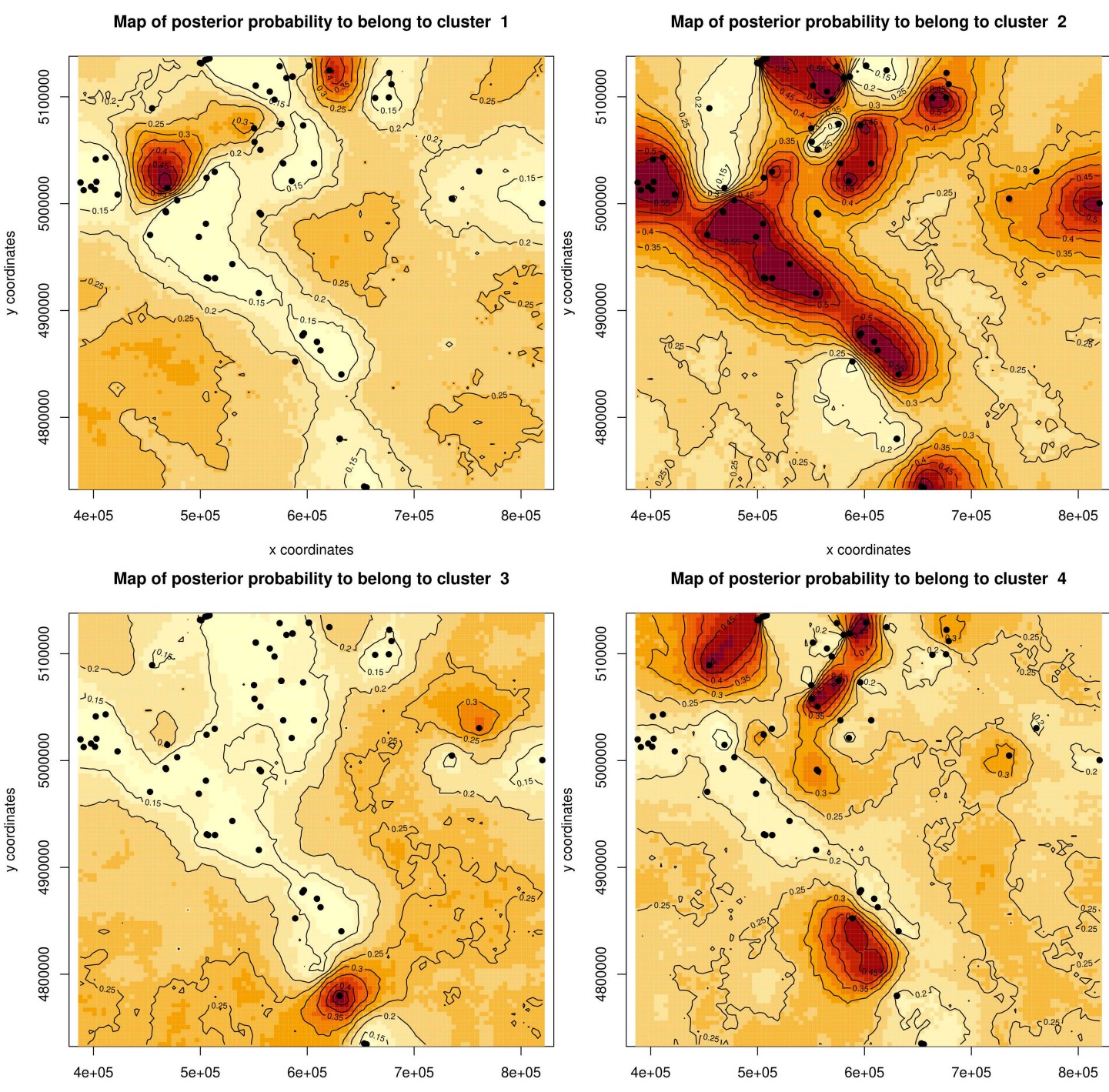

**Fig 4. Map showing spatially explicit maps of posterior probabilities for assignment to each of the four inferred genetic clusters, based on Geneland analysis using an independent allele frequencies model that accounts for potential null alleles.** The analysis included samples from Croatia and Slovenia (represented by black points), with darker areas indicating higher probabilities of cluster membership. The Adriatic coast is marked by a blue line.

**Table 2. Pairwise F$_{ST}$ between clusters/populations, as defined in Fig 3.**

|  | Hvar | Krk | Lastovo | Mainland HR |
|---|---|---|---|---|
| Krk | **0.442** |  |  |  |
| Lastovo | **0.546** | 0.145 |  |  |
| Mainland HR | **0.264** | 0.0150 | **0.188** |  |
| Mainland SI | **0.312** | 0.0354 | **0.219** | 0.005 |

* Values in bold indicate significant differences with alpha=0.05.

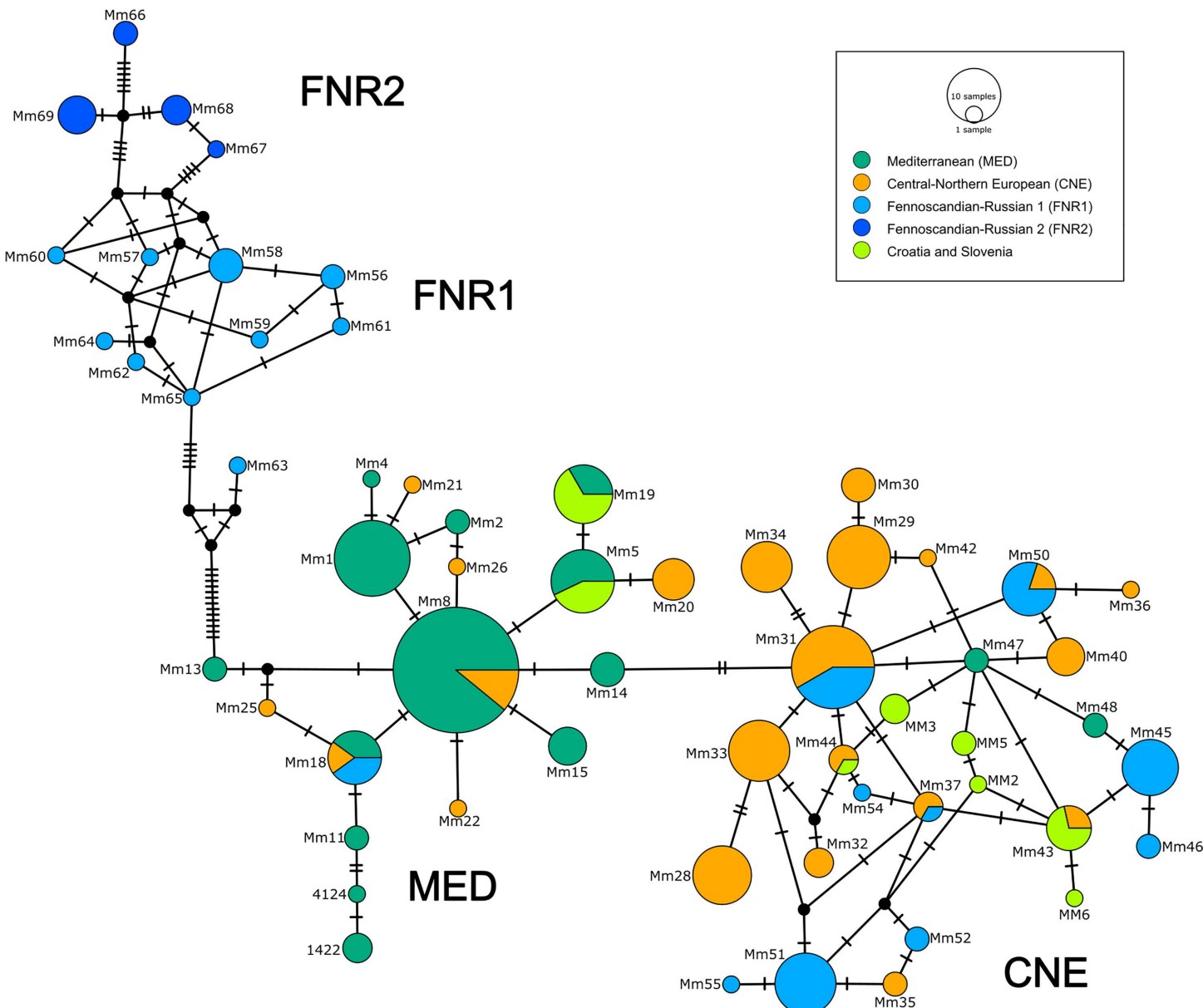

**Fig 5. The median-joining network of mtDNA CR haplotypes from the European pine marten populations is colour-coded according to phylogroups.** The number of mutations (greater than one) between haplotypes is represented by perpendicular lines. Circle size is proportional to the frequency of each haplotype. For haplotype designations and distributions, see S2 Table.

Molecular diversity indices were calculated using the haplotype sequences from this study and previously published data (Tables 3 and S2). Overall haplotype diversity (Hd) was high at 0.935±0.008, with nucleotide diversity (π) of 0.0135±0.001. Among the three phylogeographic groups, the Central–Northern European group showed high haplotype diversity (0.932±0.008) but relatively low nucleotide diversity (0.007±0.001). The Fennoscandian–Russian group had similarly high haplotype diversity (0.930±0.030) with a slightly higher nucleotide diversity (0.012±0.001). In contrast, the Mediterranean group exhibited the lowest values for haplotype and nucleotide diversity, at 0.770±0.031 and 0.003±0.001, respectively.

Tajima's D values were negative in all groups except the FNR group, which showed a positive but non-significant value (0.568, p>0.10). Although none of the Tajima's D values reached the threshold of statistical significance, the consistently negative values in the overall dataset and in the CNE and MED groups may indicate potential population expansion. This interpretation is further supported by the significantly negative Fu's Fs values observed in the overall dataset (−19.370) as well as in CNE (−9.992) and MED (−9.939) groups.

## Discussion

This study provides new insights into the genetic diversity and phylogeographic structure of two sympatric mustelids, i.e., pine marten and stone marten, across a biogeographically complex region spanning Slovenia and Croatia. Using complementary mitochondrial and microsatellite markers, we assessed population structure at multiple spatial scales and evaluated signatures of demographic change. The combined evidence highlights how postglacial range dynamics, regional barriers, and insular–mainland separation have shaped contemporary genetic patterns in both species, while also revealed species-specific differences in lineage distribution and connectivity.

### Mitochondrial diversity and phylogeography

The mitochondrial haplotype network identified two well-supported stone marten phylogenetic groups: one aligning with the widely distributed Major Eurasian lineage, and the other that clusters with haplotypes from eastern Turkey. This division is congruent with the mitochondrial phylogeny previously reported by Arslan et al. [27], who described two distinct clades: one restricted to eastern Turkey and another encompassing western Turkey, Greece, and Bulgaria. Within our analysis, haplotypes belonging to the Major Eurasian group were detected across both Croatia and Slovenia, whereas haplotypes associated with the eastern Turkish lineage (MF6, MF8, MF9, and MF10) were found only in southern Croatia, primarily in coastal and island populations. Additional eastern Turkish haplotypes shown in Fig 2 (e.g., MF_H5) derived from GenBank reference sequences (Northern and Eastern Turkey) and were not observed in Slovenia. The co-occurrence of the widespread Major Eurasian lineage and the geographically restricted eastern Turkish lineage in southern Croatia is consistent with localised lineage overlap, potentially reflecting secondary contact between lineages of different refugial origins in the northern Balkans. The grouping of these haplotypes with eastern Anatolian lineages is consistent with findings by Ishii et al. [24], who also proposed the existence of multiple glacial refugia in Anatolia and the Balkans that contributed to the current diversity and distribution of stone marten in Europe. The occurrence of widely

**Table 3. Molecular diversity indices based on mitochondrial control region sequences for pine marten.**

|  | P | h | Hd (SD) | π (SD) | Tajima's D | Fu's Fs |
|---|---|---|---|---|---|---|
| Overall (n=318) | 45 | 55 | 0.935 (0.008) | 0.0135 (0.001) | −0.224 (p>0.10) | −19.370 (p=0.003) |
| CNE (n=152) | 20 | 27 | 0.932 (0.008) | 0.007 (0.001) | −0.240 (p>0.10) | −9.992 (p=0.007) |
| FNR (n=25dn) | 19 | 14 | 0.930 (0.030) | 0.012 (0.001) | 0.568 (p>0.10) | −2.664 (p=0.132) |
| MED (n=141) | 16 | 18 | 0.770 (0.031) | 0.003 (0.001) | −1.350 (p>0.10) | −9.939 (p=0.001) |

P: number of polymorphic sites, h: number of haplotypes, Hd: haplotype diversity, π: nucleotide diversity, SD: standard deviation.

distributed haplotypes (e.g., MF_H26 and MF-1), which are also reported across broad parts of Europe, is consistent with postglacial expansion and historical connectivity among populations. This pattern also highlights Turkey as an important refugium for the stone marten during the Last Glacial Maximum [22]. Consistent with the geographic distribution of mtDNA lineages (Fig 2), our results support postglacial range expansion of the Major Eurasian lineage across southeastern and central Europe, whereas Turkish lineages indicate persistence in Anatolia during the LGM [24]. It is assumed that populations from the western Turkish refugium migrated to central and western Europe, but also to south-eastern European countries (e.g., Greece, Bulgaria), in the postglacial period. Two mtDNA haplogroups were recovered: one clustering mainly with eastern Turkish lineages, and the other corresponding to the widespread Major Eurasian group. Unlike the marked east–west split reported for Turkey, the Major Eurasian group shows little geographic structuring across Europe and adjacent regions [24,27]. This pattern suggests historical gene flow across the species' Eurasian range and supports the idea that contemporary populations in the northern Adriatic region belong to a widely distributed mitochondrial lineage.

Such patterns of multiple refugial origins, complex phylogeographic structures, and range expansions have also been documented in other mustelids, including European badger (*Meles meles*) [50], polecat (*Mustela putorius*) [51], least weasel (*Mustela nivalis*) [52,53], and pine marten [29]. All studies reinforce the idea that south-eastern Europe serves as a complex contact zone where divergent lineages of several mustelid species can coexist and potentially hybridise. Taken together, the pairwise $F_{ST}$, Geneland and PCA analyses reveal a consistent pattern of geographic population structure, with island populations showing strong genetic differentiation and mainland populations forming broader, less distinct clusters. These findings suggest limited gene flow between island and mainland populations and historical isolation of stone martens inhabiting remote islands.

The pine marten haplotypes from Croatia and Slovenia correspond to the main haplogroups defined by Ruiz-González et al. [2] and confirm the presence of two main mitochondrial lineages: the Mediterranean and the Central–Northern European clades. All novel haplotypes detected in our study fall within the CNE group, supporting their broad postglacial expansion and revealing new, previously unreported haplotype variants in the Alpine and Dinaric regions. Following the topology of the phylogenetic tree of Ruiz-González et al. [29], the clustering of the pine marten samples reflects common ancestry with populations from Central and Eastern Europe, rather than the more differentiated Mediterranean clade.

## Demographic trends and historical processes

The strongly negative Fu's Fs values detected for both stone and pine marten (particularly in the CNE clade for pine marten and in the Major Eurasian group for stone marten) coincide with recent population expansion after the Last Glacial Maximum. The FNR group, however, showed a smaller and non-significant Fu's Fs value (−2.664), suggesting a weaker or absent signal of expansion in this lineage. These findings are consistent with patterns previously reported by Ruiz-González et al., who also identified signals of demographic expansion and historical structuring among *M. martes* mitochondrial lineages across Europe [29]. Although Tajima's D values were not significant, the combination of negative Fu's Fs, high haplotype diversity, and star-like haplotype network structures suggests population growth following the Last Glacial Maximum. In stone marten, the combination of high haplotype diversity and relatively low nucleotide diversity together with strongly negative and significant Fu's Fs values supports the hypothesis of a recent demographic expansion, particularly within the Major Eurasian haplogroup, as also suggested by Arslan et al. [24] and Ishii et al. [27]. Our results therefore support the view that populations from southern refugial areas in the Balkans and Anatolia likely contributed to the postglacial recolonization of central and western Europe, as also inferred from broader phylogeographic studies of both marten species.

## Population structure and spatial differentiation of stone marten

Microsatellite data revealed a moderate genetic structuring in stone marten, with distinct clusters corresponding to island populations (e.g., Hvar) and more admixed mainland groups. The PCA analysis showed a strong separation of insular

populations along the first principal component, while individuals from the Croatian and Slovenian mainland formed overlapping clusters, suggesting continuous gene flow and limited barriers to dispersal. The clearest structuring reflects island–mainland contrasts, consistent with the sea acting as the principal barrier in our study area. This pattern aligns with other studies showing that natural geographical barriers and habitat discontinuities can shape genetic structuring in mesocarnivores [20,54,55]. The Geneland spatial analysis further confirmed the presence of four genetic clusters with geographical coherence. While one cluster was clearly associated with the southern Adriatic region (island of Hvar), others showed a broader distribution across central and northern areas. This pattern indicates both localized genetic drift in isolated populations and broad-scale gene flow between connected regions. Remarkably, the Dinaric Mountains, traditionally considered a geographical barrier, have not disrupted the genetic connectivity of stone martens, but may even have facilitated their movement, as has been observed in other mesocarnivores in alpine and subalpine habitats. These differentiation patterns are consistent with the findings of Wereszczuk et al. [21], who reported a moderate population structure in stone marten across central-eastern Europe. Their study revealed significant $F_{ST}$ values between regions in Poland and eastern Germany, likely reflecting historical fragmentation and current habitat discontinuities. Although our $F_{ST}$ values are similarly moderate, the geographical structuring differs. While Wereszczuk et al. identified a broader east–west division and reduced connectivity within agricultural landscapes, our data revealed fine-scale differentiation between islands and mainland sites, emphasising the natural isolation of island populations and relatively unrestricted gene flow across the rugged mainland terrain [21]. Collectively, these results emphasize the flexibility of stone marten in navigating heterogeneous landscapes, even though local population structures may evolve through geographical or ecological isolation. In addition, the Mantel test showed that isolation by distance was not statistically significant (p = 0.15), suggesting that geographical distance alone cannot explain the observed genetic differentiation in the populations studied. This further supports the influence of landscape features, such as insularity or habitat discontinuities, over purely spatial separation.

Although our results provide valuable insights into the genetic diversity and structure of martens in the northern Dinaric and Adriatic regions, several limitations must be acknowledged. The most notable is the limited sample size for pine martens, particularly in Slovenia, which restricts the power of inferences regarding phylogeographic structure and demographic history. Furthermore, the use of only a partial mitochondrial marker and a relatively small set of microsatellite loci, while informative, does not capture the full genomic complexity of these species. Future research should therefore apply genome-wide approaches, such as whole-genome resequencing, to obtain a higher-resolution picture of population connectivity, local adaptation, and evolutionary history. Such genomic tools will be especially important for resolving fine-scale structure, detecting subtle admixture, and understanding the functional basis of adaptation in response to both natural and anthropogenic pressures. Integrating ecological data with genomic analyses will also enhance our ability to inform conservation and management strategies for these sympatric carnivores in a changing landscape.

## Conclusions

This study provides new insights into the genetic structure and phylogeographic history of stone and pine martens in south-eastern Europe, a region of recognized biogeographic complexity. By analysing mitochondrial DNA and microsatellites, we identified significant genetic diversity and distinct lineage patterns shaped by postglacial expansion and historical isolation. The presence of both widespread and regionally restricted haplotypes underscores the role of the Balkans (and Anatolia, as revealed earlier by other authors) as important glacial refugia. Microsatellite data revealed a moderate population structure of the stone marten, particularly highlighting the genetic distinctiveness of island populations in Croatia.

However, overall connectivity between mainland populations remains high, and the lack of significant isolation by distance suggests that geographical barriers such as the Dinaric Mountains do not substantially limit gene flow in this species.

Despite these new findings, our study also highlights the need for more comprehensive genomic approaches to better resolve population history, admixture, and adaptive variation. Future studies should incorporate genome-wide markers

and integrate environmental data to deepen our understanding of the evolutionary and ecological processes shaping marten populations in this heterogeneous region.

## Supporting information

**S1 Fig. Haplotype accumulation curves.** Haplotype accumulation curves by population of stone marten (*Martes foina*) samples from the present study from Croatia (red line) and Slovenia (cyan line).
(TIF)

**S2 Fig. Genetic and geographical distances.** Relationship between genetic and geographic distances in *Martes foina* based on microsatellite data. The colour gradient represents the local density of data points (from light yellow = low density to dark blue = high density) obtained by two-dimensional kernel density estimation. No significant isolation by distance (IBD) was detected (Mantel test, $p = 0.133$); therefore, no regression line is shown. Dgeo – geographical distance (degrees); Dgen – genetic distance.
(TIFF)

**S1 Table. Sample data.** Basic data on pine marten (*Martes martes*) and stone marten (*Martes foina*) analysed samples included in the study. The "Haplotype" column represents the recognized haplotype, the "Haplotype – network" column represents how the haplotypes are labelled in the haplotype networks (Figs 2 and 6).
(DOCX)

**S2 Table. *Martes foina* haplotypes with GenBank accession numbers.** The "Haplotype - GenBank" column represents the haplotype names obtained from GenBank, the "Haplotype – network" column represents how the haplotypes are labelled in the haplotype network (Fig 2).
(DOCX)

**S3 Table. *Martes martes* haplotypes with GenBank accession numbers.** The "Haplotype - GenBank" column represents the haplotype names obtained from GenBank, the "Haplotype – network" column represents how the haplotypes are labelled in the haplotype network (Fig 2).
(DOCX)

**S4 Table. Microsatellite loci primers.** Microsatellite loci analysed in fragment analysis of pine marten (*M. Martes*) and stone marten (*M. foina*) published by Basto et al. 2010.
(DOCX)

**S5 Table. Genetic variation in the analysed samples of stone marten.**
(DOCX)

## Acknowledgments

We thank all the collaborators and founders who help accelerate collaboration with hunters as citizen scientists in wildlife monitoring and biodiversity research. We thank many hunters, foresters and other colleagues for collecting and providing the samples.

## Author contributions

**Conceptualization:** Elena Buzan.

**Formal analysis:** Luka Duniš, Tilen Komel.

**Funding acquisition:** Elena Buzan, Boštjan Pokorny, Magda Sindičić.

**Methodology:** Elena Buzan, Luka Duniš, Tilen Komel.

**Project administration:** Elena Buzan.

**Resources:** Boštjan Pokorny, Zoran Marčić, Magda Sindičić.

**Supervision:** Elena Buzan.

**Validation:** Elena Buzan, Luka Duniš, Tilen Komel.

**Visualization:** Luka Duniš.

**Writing – original draft:** Elena Buzan.

**Writing – review & editing:** Elena Buzan, Luka Duniš, Tilen Komel, Boštjan Pokorny, Carlos Rodríguez Fernandes, Magda Sindičić.

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
