## [Decision Letter · Decision Letter 0]

23 Sep 2025

Dear Dr. Buzan,

Thank you for submitting your manuscript to PLOS ONE. After careful consideration, we feel that it has merit but does not fully meet PLOS ONE’s publication criteria as it currently stands. Therefore, we invite you to submit a revised version of the manuscript that addresses the points raised during the review process.

Both reviewers see merit in the data, though they feel the writing, framing and discussion needs more work.

The main points are the following.

Conduct analyses that are blind to country designation, PCA instead of DAPC like reviewer 2 recommends. Reviewer 1 also questions this subdivision.

Provide broader introduction, why does studying these two species and their diversity matter?

Discuss results for both species.

Submit the sequence data into genebank.

Improve the text. It was described as choppy and lacking flow. Urge the senior authors to help the junior scientists here to improve the manuscript and their writing skills. The reviewers offer many good suggestions, but the journal does not provide copy editing, and this has to come from the submitting authors.

Intermediate points

Figure 1 is truncated. Does it show sampling for one or both species? See also reviewer 1 on figure presentation.

We look forward to receiving your revised manuscript.

Kind regards,

Arnar Palsson, Ph.D.

Academic Editor

PLOS ONE

“This study was funded by: (i) the Slovenian Research and Innovation Agency (programme groups P1–0386 and P4–0107), (ii) the STEPCHANGE European Union's Horizon 2020 Research and Innovation Program under grant agreement No. 101006386, iii) The PRO COAST European Union's Horizon Europe Research and Innovation Program under grant agreement No. 101082327.”

Please state what role the funders took in the study.  If the funders had no role, please state: "The funders had no role in study design, data collection and analysis, decision to publish, or preparation of the manuscript"

4. Please note that your Data Availability Statement is currently missing a direct link to access each database. If your manuscript is accepted for publication, you will be asked to provide these details on a very short timeline. We therefore suggest that you provide this information now, though we will not hold up the peer review process if you are unable.

5. We note that Figures 1 and 3 in your submission contain [map/satellite] images which may be copyrighted. All PLOS content is published under the Creative Commons Attribution License (CC BY 4.0), which means that the manuscript, images, and Supporting Information files will be freely available online, and any third party is permitted to access, download, copy, distribute, and use these materials in any way, even commercially, with proper attribution. For these reasons, we cannot publish previously copyrighted maps or satellite images created using proprietary data, such as Google software (Google Maps, Street View, and Earth). For more information, see our copyright guidelines: http://journals.plos.org/plosone/s/licenses-and-copyright.

1. You may seek permission from the original copyright holder of Figures 1 and 3 to publish the content specifically under the CC BY 4.0 license.

6. We note that there is identifying data in the Supporting Tables S1, S2 and S3. Due to the inclusion of these potentially identifying data, we have removed this file from your file inventory. Prior to sharing human research participant data, authors should consult with an ethics committee to ensure data are shared in accordance with participant consent and all applicable local laws.

-Location data

Please remove or anonymize all personal, ensure that the data shared are in accordance with participant consent, and re-upload a fully anonymized data set. Please note that spreadsheet columns with personal information must be removed and not hidden as all hidden columns will appear in the published file.

Additional Editor Comments:

Both reviewers see merit in the data, though they feel the writing, framing and discussion needs more work.

The main points are the following.

Conduct analyses that are blind to country designation, PCA instead of DAPC like reviewer 2 recommends. Reviewer 1 also questions this subdivision.

Provide broader introduction, why does studying these two species and their diversity matter?

Discuss results for both species.

Submit the sequence data into genebank.

Improve the text. It was described as choppy and lacking flow. Urge the senior authors to help the junior scientists here to improve the manuscript and their writing skills. The reviewers offer many good suggestions, but the journal does not provide copy editing, and this has to come from the submitting authors.

Intermediate points

Figure 1 is truncated. Does it show sampling for one or both species? See also reviewer 1 on figure presentation.

Reviewers' comments:

Reviewer's Responses to Questions

**Comments to the Author**

1. Is the manuscript technically sound, and do the data support the conclusions?

Reviewer #1: Partly

Reviewer #2: Partly

2. Has the statistical analysis been performed appropriately and rigorously?

Reviewer #1: Yes

Reviewer #2: No

3. Have the authors made all data underlying the findings in their manuscript fully available?

Reviewer #1: No

Reviewer #2: Yes

4. Is the manuscript presented in an intelligible fashion and written in standard English?

Reviewer #1: Yes

Reviewer #2: Yes

Reviewer #1: The manuscript entitled: “First insight into genetic diversity of two sympatric marten species between the Alps and Adriatic islands” contains original and valuable data and results concerning genetic diversity of two species of martens in Slovenia and Croatia. Although the sample size is not large (especially in case of pine martens), the study is properly designed, the methods and analyses are correct. However, the manuscript requires major revision before publication. First of all, the text should be revised and shortened and repetitions should be removed. Not all analyses are described in Methods. Also presentation of some results needs some improvement. Furthermore, interpretation of some results should be revised.

Detailed comments

The provided information concerning number of analysed samples is different in the Abstract (29 pine martens +182 stone martens), Methods (29 pine martens + 199 stone martens) and Results (107 stone martens + 29 pine martens). This information should be revised and only the number of samples for which the results have been obtained should be reported in the Abstract.

The attached figures should be numbered.

Fig. 1 should be better described: e.g. what does the colour of samples mean? In my version of the Fig. 1 the map is cut in its northern parts, so some samples are not fully visible and there is a fragment of insert, which is also not visible. I think the map should have a legend, a scale and the northern direction marked. Also names of some geographic localities mention in the text as e.g. Hvar Island, Krk, Lastovo Islands should be marked on the map. Futhermore, a small insert with contours of Europe with localization of your study area would be helpful, especially for non-European readers.

Fig. 5 could be moved to the Supplementary (it is enough if the lack of IBD is mentioned in the text). If IBD is not statistically significant, the line should be removed from the Figure. Moreover, add information, what does the colour gradient on the figure mean.

Introduction should be shortened and focus mainly on results of previous studies, which are closely related to the aims of the present study (e.g. the text in lines 71-88 should be reduced). Also the detailed information concerning the number of road kills and huntig bag could be omitted.

Line 59 there is a lack of something in this sentence (probably before “found”).

Line 113 a lack of point before “Due”

Line 219 This sentence is unclear. I would add “in 7 analysed samples” after “four”.

I think it would be better if you present the results of mtDNA diversity analyses for the whole dataset for the stone martens without dividing it into Slovenian and Croatian samples. You have only 7 Slovenian samples and 100 Croatian, so there is no sense to report and compare the values of genetic diversity indicators for those two groups of samples, especially as these two mainland populations are genetically similar.

Table S1: Are the all accession numbers for your sequences provided in Table S2?

Lines 242-245: this fragment should be moved to the Discussion part

Lines 262-275 This fragment of the text contains numbers (values of indices), which have been already presented in the Table 2. This is unnecessary repetition, the numbers should be removed from the text. There is also no need to described in details the Tajima's D indices, if they are not statistically significant (so the sentence in lines 265-267 could be deleted).

Analyses of Taijma’s D and Fu’sFs are not described in Methods.

The presentation of the DAPC results should be improved. It is not clear, which of your analysed samples belong to which DAPC cluster. Maybe you could marked on the map (using the same colours as in the DAPC figure) the distribution of samples belonging to each DAPC cluster. You described the clusters in the text but the geographic names mentioned in the text do not match with the numbers and colours of the clusters. Also spatial PCA (sPCA) could be useful to present the spatial distibution of the genetic clusters. You presented the spatial genetic structure obtained in Geneland but the number of clusters obtained in those two analyses are not equal, so it will be good to compare results of both approaches on the maps.

Lines 321-324: This fragment should be moved to Discussion.

Lines 368 – 380: This fragment should be moved to Discussion.

Line 409. It is not clear. Do you mean here western refugium in Turkey?

Lines 419-420 It is not clear what do you mean by: “suggesting previously undocumented genetic variation in the Alps and the Dinaric region”. Please, revise the sentence.

Lines 29, 390, 438-439 You did not analyse anthropogenic landscapes or barriers in your study. According to the obtained results the most genetically distinct populations inhabit islands, so its seems that geographical barriers such as the sea play more important role in shaping the genetic structure of martens in your study area than the anthropogenic barriers (at least for the stone martens, as in case of pine martens you analysed only mtDNA of 28 individuals and the results of spatial genetic analyses are not presented, so it is difficult to asses the gene flow in the population). The text should be revised.

Authors declared that all data are fully available without restriction but I have not found any information concerning the access to the microsatellite dataset and the Gene Bank numbers of sequences obtained in this study.

Reviewer #2: In this manuscript, the authors use mtDNA and nuclear markers to investigate how two mesocarnivores, the pine marten (Martes martes) and the stone marten (Martes foina), differ in their genetic structure, diversity, and connectivity across Croatia and Slovenia. They assess genetic patterns of differentiation in ~190 stone marten and 28 pine marten individuals sampled from mainland and island populations. The results seem to reveal sub-structuring in the stome marten, mainly between island and mainland populations, while pine marten sub-structuring remains unclear.

While I think this paper has a good starting dataset and a lot of potential, I have several large comments that would need to be addressed before submission, mainly regarding the choice of groupings for the methods as well as the methods themselves. Additionally, the manuscript currently reads very choppy and is confusing and lengthy as a reader. All of this is addressed below.

Comments:

Abstract: the abstract claims “For pine marten, we found a significant genetic structuring, with pronounced differentiation between island and mainland populations, and a further substructure within the mainland. No significant isolation by distance was detected (Mantel test, p = 0.15), suggesting that genetic differentiation is driven more by habitat discontinuities and anthropogenic barriers rather than geographical distance alone. In contrast, stone marten exhibited weak genetic structure and high genetic diversity, indicating gene flow and potential landscape permeability for this more synanthropic species. These contrasting pattern underscore species-specific responses to landscape fragmentation and highlight the need to tailor management strategies accordingly.” But the structuring between island and mainland is in the stone marten for the results and discussion? Also in the discussion there is no follow-up on the pine marten distribution, etc. which is them confusing to add to the abstract. This abstract needs to be re-written to follow what is discussed in the manuscript.

Introduction

I think there is a lot of good information in the introduction, I just think it needs to be re-written substantially. I think a broader importance of looking at these two species should be highlighted first in the introduction before narrowing the focus. The authors mention that the stone marten is more ‘adaptable’ and the pine marten maybe more ‘sensitive’ to human change etc. I think this is a better set-up to the paper than it currently has. What can we learn about these two species that have different requirements. How do these different behaviors/requirements impact the degree of genetic differentiation across populations, etc. Would we expect to see differences? We would certainly expect to see differences sooner in the pine marten if there is substantial human impact. As I am not an expert in this species, I think the authors would benefit from considering how their study can inform wider marten research and connect their study to a broader viewpoint. Additionally, there is a lot of information on other studies, which is great, but as a reader it feels as if it’s a lot of information without a clear thread that leads to why this study in Croatia and Slovenia is necessary and how it contributes something novel. The authors should think to make it more streamlined, where a reader without a history in marten research can follow along.

Line 39: “spatial genetic distribution in sympatry”. I think this should be expressed differently since sympatry inherently implies overlapping spatial ranges and genetic distribution, I am not sure what is meant by this.

Line 104-110: These few lines are a bit confusing. I would restate as:

Between 2014-2024, the annual harvest of stone marten ranged from 790 (2020) to 995 (2015) individuals, and pine marten between 82 (2024) and 127 (2017). During this same period, registered roadkill ranged from 342 and 24 (2022) to 430 and 49 (2019) for stone and pine marten, respectively.

Methods

My main issue is that several of the analyses are they are based on arbitrary country assignments, which in the introduction, the authors state there are no clear barriers since there are large corridors of uninterrupted forested land.

Additionally, I think a PCA should be used instead of a DAPC, since this is not biased by sampling locations. A PCA analysis should also be carried out on the pine marten, even with low sample size, because a PCA makes no a priori assumptions on clusters, it is still an option to run this here. Clusters should be chosen after running PCA and defining clusters. Additionally, the authors should look at how these genetic clusters translate to spatial groups and report on this.

Results

General comment: the results are currently extremely long and un-structured. I would only keep what is necessary to convey the main points that are presented in the tables and figures as currently, it feels repetitive and long for what is actually being presented in the figures/tables.

202-203: why did some samples fail?

206: It would be nice here to remind the reader of how many stone marten individuals you have per country.

216-222: here the authors talk about differences between Slovenia and Croatia, but previously in the introduction talk about how the habitat is very well connected in this region. Therefore, it seems to me that a better way to define populations is by known barriers and or performing a PCA on the genetic data to see which clusters appear.

Line 220-222: These sentences are not understandable as a reader

Figure 1: I believe the map was cut off from my view, but in general, it would be good to add a background to the spatial map of forest cover as this was mentioned as being important for dispersal/movement through habitat.

Figure 2: colors between certain groups are too similar to make clear distinctions between them

Line 280-289 and Figure 3: There are numbered clusters in the figured which are described no where in the text or figure caption. I disagree with the use of the DAPC, and would suggest a normal PCA, but it does seem that there are at least 3 clear genetic clusters from their results. Which it is unclear if this falls along the Croatia/Slovenia sampling line since there is no description or explanation. However, I would suggest that the authors use these clusters to define groups to run different analyses on and also look at how the groups resulting from PCA are spatially structured. I also generally think the suthors should take the time to make the figures outside of the programs running the analysis to have a better looking more concise figure. Figure 1,2 and 3 could be combined into one figure where the clusters from a PCA can be mapped onto the sampling map and then linked to the mtDNA and nuclear DNA. See Lucena-Perez et al. 2020 for an example. Spending the time to do this greatly enhances reader comprehension and translates better the important results. As it reads currently, I as a reader am trying to find the important points from the results.

Table 3. Finally, the clusters identified are used to look at FST, which does show clear differences. It is therefore clear that using the Croatia/Slovenia split is simply not informative and unnecessary.

Line 321-324: seems more like discussion

Figure 5. unnecessary

Line 326-330: this does not need its own sub-section, this can be combined with the above results as a secondary test to FST etc.

Line 333: I think the results from stone and pine marten should be presented together, as the analysis is a similar framework and it makes the results extremely long to keep them separate. Again here, I suggest PCA results to be done first to define clusters and all other analyses be done based on those identified clusters.

Line 376-380: this is primarily discussion talking about hypothesis of expansion, etc.

Discussion:

Line 393-402: my general question here, is how are these mtDNA haplotypes distributed in space. It is clear you have samples belonging to the Eurasian lineageand from eastern turkey. Are they distributed differently in space? Are they mixed? This might give some insight into how these two lineages meet in this space.

Line 407: “there is evidence for to the existence”

Line 424-431: I am confused how if the Tajimas D were not significant the mtDNA diversity can be enoguht for demographic growth scenarios. In what time frame? Since the LGM? This seems quite evident in most species in Europe, so then the question is not, did they expand, but from where did they expand? I would be a bit more clear on what exactly from the results you are using to support you claims, and also be more clear with what questions you want to and can answer from your data.

Line 435-440: so if it is known already there are no barriers between countries, this calls into question why half of the analyses are carried out using the countries separately?

Line 449-454: This is a great summary of what is going on. I think the methods and results should follow this framework of trying to identify where there may be breaks in the gene flow between groups – and quantify how much.

Then pine martens are completely left out of the rest of the discussion. I think there may still be valuable insight for a first look into connectivity, specifically if a PCA is carried out. And it can be discussed, even with limitations, that the first look shows X, but more samples and follow-up would be needed to determine if there are distinct clusters or groups and if it is experiencing fragmentation as it is the more sensitive species.

.

Reviewer #1: No

Reviewer #2: No

---

## [Author Response · Author response to Decision Letter 1]

19 Dec 2025

We thank the Editor and Reviewers for their valuable comments and suggestions. We have carefully addressed all points in detail. The full Response to Editor and Response to Reviewers documents have been uploaded as separate supporting files with this submission.

---

## [Decision Letter · Decision Letter 1]

9 Feb 2026

Dear Dr. Buzan,

Thank you for submitting your manuscript to PLOS ONE. After careful consideration, we feel that it has merit but does not fully meet PLOS ONE’s publication criteria as it currently stands. Therefore, we invite you to submit a revised version of the manuscript that addresses the points raised during the review process.

We look forward to receiving your revised manuscript.

Kind regards,

Arnar Palsson, Ph.D.

Academic Editor

PLOS One

Journal Requirements:

Additional Editor Comments:

The manuscript is much improved. R1 has several text corrections, please make those and resubmit promptly.

Reviewers' comments:

Reviewer's Responses to Questions

**Comments to the Author**

Reviewer #1: (No Response)

Reviewer #2: All comments have been addressed

2. Is the manuscript technically sound, and do the data support the conclusions?

Reviewer #1: (No Response)

Reviewer #2: Yes

3. Has the statistical analysis been performed appropriately and rigorously?

Reviewer #1: (No Response)

Reviewer #2: Yes

4. Have the authors made all data underlying the findings in their manuscript fully available?

Reviewer #1: (No Response)

Reviewer #2: Yes

5. Is the manuscript presented in an intelligible fashion and written in standard English?

Reviewer #1: (No Response)

Reviewer #2: Yes

Reviewer #1: The manuscript has been significantly improved but it still requires some further revision as some parts of the text (especially in Results and in Discussion), some of the tables and figures are not clear.

In this revsion the line numbers correspond to the tracking change version of the manuscript.

Abstract

Lines 20-21: I would rather not write that you used “landscape genetic approach”. You did not compare the genetic differentiation indices with landscape structure. The spatial analyses are limited. Your rather performed population genetic studies here.

Introduction

Line 42

Add “their” after “shape”

Add “among populations” after “connectivity”

Line 43: add “examples of” after “are”

Line 89: I suppose you mean here rather glacial than postglacial refugia

Line 120: add “connectivity among populations of the species” after “that supports”

Methods

Line 152: Do you mean here collected samples or included in the final analyses? This should be clarified.

Results

Lines 239-241 I would move this once sentence paragraph to Methods or merge it with the next paragraph (and subchapter). Otherwise you have once sentence subchapter.

While I recognise the importance of the information on the genetic diversity of stone martens in Slovenian and Croatian populations for the management of these species in these countries, I still believe that Table 1 and all information on the genetic indices of Slovenian and Croatian martens should be moved to the Supplementary Materials (or published in a separate paper in a journal for wildlife managers). This division of the population does not provide any biological insight, especially given the unequal sample sizes (7 individuals from Slovenia and 97 from Croatia), and it decreases the scientific value of the manuscript.

Line 246: How many previously published haplotypes?

Line 260 remove the full stop before “(n = 7)”

Line 291: Which part of Russia and Ukraine do you mean here? Those are very large countries and Russia covers not only large part of Europe but also Asia.

Line 278: Are these haplotypes from your study or from previously published studies? It should be clearly written here.

Explanation to Table 2: Provide here information what is the origin of the data you present in this Table. Are they both the sequences obtained in your study and available in GenBank?

Line 249: You wrote here “No novel mtDNA haplotypes were detected in either Croatia and Slovenia” and in line 286 you wrote “… novel haplotypes from Croatia…”. Please clarify this mismatch between these two sentences.

Line 278: You wrote here that 89 haplotypes were included in the network but in Table 2 you provided information that there were 47 haplotypes and 299 samples. This mismatch should also be explained.

Fig 2: Add here a lable which group is Eurasian and which is Eastern Turkey (similarly as in Fig. 4).

Fig S2: I see only internal part of the figures, the x and y axes and explanations to them are not visible.

Explanation to S1 Table: add “samples” after “(Martes foina)” Clarify here also what do you mean here by “included”: “collected samples” or “analysed samples” or “collected and analysed samples”

Fig. 3 label is very difficult to read (resolution of the figure is too low) and the names of the populations should be explained in the explanation to this figure and maybe marked also in Fig. 1.

Fig. 4: The colour labels of the haplotypes need revision: now FNR1 are in CNE and Croatian and Slovenian haplotypes are both in MED and CNE. Maybe you should label the haplotypes according to geographic regions/countries as in Fig. 2.

Text in lines 441-444 and 447-448 should be moved to Discussion.

Discussion

The whole first paragraph should be revised as it is difficult to follow, is partly illogical and contains repetitions.

Lines 465 - 469: “In our dataset, haplotypes belonging to the Major Eurasian group were found across both Croatia and Slovenia, while haplotypes associated with the eastern Turkish group (MF6, MF8, MF9, and MF10) were restricted to southern Croatia, particularly in populations closer to the Adriatic coast and islands. This spatial pattern suggests a secondary contact zone in the northern Balkan region, where lineages of different refugial origins may overlap”. I think this fragment is illogical: how the presence of the Major Eurasian group in Croatia and Slovenia suggests a a secondary contact zone in the northern Balkan region, where lineages of different refugial origins may overlap?”. This text needs revision.

Lines 467-468: “restricted to southern Croatia, particularly in populations closer to the Adriatic coast and islands” – this statement is unclear for me too as according to Fig. 2 some haplotypes of the eastern Turkish group were found also in Slovenia (e.g. MF_H5).

In Lines 474-477 You wrote: “The presence of widespread haplotypes such as MF_H26 and MF-1 in Slovenia and Croatia indicates postglacial expansion and gene flow from the Balkans across the continent. This pattern also highlights Turkey as an important refugium for the stone marten during the Last Glacial Maximum [22].” I think this fragment needs revsion to be more logical (or you can just delete the second sentence). Also do you mean haplotypes widespread in large areas of Europe?

Lines 477-479: “Our results further support the idea that the refugia population on the Balkan Peninsula expanded into western Turkey, large parts of Europe, and Central Asia after the LGM, facilitated by the absence of significant geographical barriers”. I think this statement will be better support by your results if you present the distribution of different mtDNA lineages/haplotypes on a map.

Lines 483 - 489: This fragment contains information which is a repetition of the information provided earlier in this paragraph, so this text needs some revision and the whole paragraph should be shortened and written more briefly.

Line 556-557: This last sentence should be deleted as it is a repetition of the information provided earlier in the text (line 554-555).

Reviewer #2: I believe this manuscript is now ready for acceptance. While the previous version already rested on a solid foundation in terms of sampling, analyses, and overall study design, the revised manuscript represents a substantial improvement in clarity, structure, and accessibility. In particular, the revised introduction provides a much clearer and more engaging framework for the study, effectively setting up both the biological motivation and the broader relevance of the work. The research questions are now articulated in a way that is easy to follow, even for readers who may not be specialists in the specific system or methods used.

The authors have done an excellent job of explaining why this study matters, both within the context of the focal system and for the wider field. By more clearly linking the results to overarching concepts and ongoing debates, the manuscript now communicates its significance beyond a narrow audience. This improved framing allows the reader to better appreciate how the analyses contribute to current understanding and why the findings are timely and relevant.

Overall, the revisions have transformed a technically strong but previously dense manuscript into a clear, coherent, and compelling piece of work. The study is now accessible to a broader scientific audience without sacrificing rigor or depth, which greatly strengthens its impact. For these reasons, I fully support acceptance of the manuscript in its current form.

.

Reviewer #1: No

Reviewer #2: No

---

## [Author Response · Author response to Decision Letter 2]

26 Feb 2026

Dear Dr. Palsson,

Thank you for your message and for the opportunity to submit a revised version of our manuscript to PLOS ONE. We appreciate the time and effort invested by you and the reviewers in evaluating our work.

We confirm that we prepared a revised manuscript addressing all points raised during the review process.

With the resubmission, we included:

• a detailed Response to Reviewers document addressing each comment point-by-point;

• a Revised Manuscript with Track Changes highlighting all modifications; and

• a clean, unmarked Manuscript version.

Thank you again for the invitation to revise.

Kind regards,

Elena

---

## [Editor Report · Decision Letter 2]

9 Mar 2026

First insight into genetic diversity of two sympatric marten species between the Alps and Adriatic islands

PONE-D-25-40165R2

Dear Dr. Buzan,

We’re pleased to inform you that your manuscript has been judged scientifically suitable for publication and will be formally accepted for publication once it meets all outstanding technical requirements.

Kind regards,

Arnar Palsson, Ph.D.

Academic Editor

PLOS One
---

## [Editor Report · Acceptance letter]

PONE-D-25-40165R2

PLOS One

Dear Dr. Buzan,

I'm pleased to inform you that your manuscript has been deemed suitable for publication in PLOS One. Congratulations! Your manuscript is now being handed over to our production team.

Kind regards,

on behalf of

Dr. Arnar Palsson

Academic Editor

PLOS One